# MAD: Multi-Alignment MEG-to-Text Decoding

## Abstract

Deciphering language from brain activity is a crucial task in brain-computer interface (BCI) research. Non-invasive cerebral signaling techniques including electroencephalography (EEG) and magnetoencephalography (MEG) are becoming increasingly popular due to their safety and practicality, avoiding invasive electrode implantation. However, current works under-investigated three points: 1) a predominant focus on EEG with limited exploration of MEG, which provides superior signal quality; 2) poor performance on unseen text, indicating the need for models that can better generalize to diverse linguistic contexts; 3) insufficient integration of information from other modalities, which could potentially constrain our capacity to comprehensively understand the intricate dynamics of brain activity.

This study presents a novel approach for translating MEG signals into text using a speech-decoding framework with multiple alignments. Our method is the first to introduce an end-to-end multi-alignment framework for totally unseen text generation directly from MEG signals. We achieve an impressive BLEU-1 score on the *GWilliams* dataset, significantly outperforming the baseline from 5.49 to 10.44 on the BLEU-1 metric. This improvement demonstrates the advancement of our model towards real-world applications and underscores its potential in advancing BCI research.

## 1 Introduction

Decoding brain to language has emerged as a rapidly developing area of neurotechnology, offering semantic communication and control for general Brain-Computer-Interface (BCI) tasks. This region has garnered growing focus as it may profoundly impact individuals with verbal and movement disabilities resulting from conditions such as severe spinal cord trauma or end-stage amyotrophic lateral sclerosis (ALS). Moreover, the scope of brain-to-text technology extends to pioneer novel human-machine interfaces, allowing seamless control of prosthetic limbs, software, and virtual environments, shifting the paradigm of interaction for both able-bodied individuals and those with disabilities, and re-defining what is achievable in both everyday life and professional spheres.

Under this scope, various previous works have explored this area in multiple ways. Pioneer researchers first verify this idea by using invasive signals such as Electrocorticography (ECoG) Anumanchipalli et al. (2019); Wang et al. (2020); Willett et al. (2021); Wang et al. (2021). Recently, these invasive methods Willett et al. (2023); Metzger et al. (2023) concentrate on decoding speech, phonemes or letter from ECoG signals and have achieved remarkably high accuracy using limited word sets for real-time brain-to-text translation. However, these invasive-signal-based approaches pose significant medical risks and challenges for long-term use.

Non-invasive techniques, therefore, present a safer and more sustainable alternative, albeit with their own set of challenges. Wang et al. Wang & Ji (2022) showcased a method for translating EEG signals into text with an extensive lexicon, utilizing language models that had been pretrained on EEG data features at word-level. Duan et al. Duan et al. (2023) progressed this methodology by interpreting raw EEG signals directly, devoid of reliance on temporal indicators, but their models still relied heavily on teacher-forcing for evaluation, limiting their ability to generate meaningful sentences autonomously in real-life scenarios. At the same time, although Magnetoencephalography (MEG) provides better signal quality, previous works Dash et al. (2020); Csaky et al. (2023); Ghazaryan et al. (2023) on MEG have primarily focused on decoding limited classes or short phrases from MEG signals, showing limited success in generating whole sentences and complete semantic segments.

Furthermore, as pointed out by Jo et al. Hyejeong et al. (2024), all previous works in EEG-to-Text translation following Wang's method Wang & Ji (2022) meets the "decoder dominated" problem. It means that given a strong decoder and noisy EEG input, these models are more likely to memorize the text distribution corresponding to certain statistical features rather than mapping EEG to semantic texts. Thus, these models have similar performances even when we replace EEG input with random noise. Besides, due to the nature of limited data and the non-understandability of the neural signal, it is difficult to train and evaluate the model. Yang et al. Yang et al. (2024) proposed NeuSpeech model on MEG to text task, however, their model is evaluated on the text that is seen in the training set, which does not meet the need for open-vocabulary translation. Defossez et al. Défossez et al. (2023) highlighted the potential to decode speech perception from MEG signals, where they matched MEG signals with corresponding speech segments. However, their approach was limited to classification tasks and could not generate sentences directly from MEG signals. This underscores a significant gap in the current state of MEG-based brain-to-text decoding.

In this paper, we propose an end-to-end framework for open-vocabulary MEG-to-Text translation capable of generalized on unseen text for real-life utilization. However, as mentioned by Yang et al. Yang et al. (2024), relying solely on traditional text loss, as done in previous works, is inadequate in unseen text scenarios. Our intuition is that incorporating additional information from different modalities can enhance the model's performance. Therefore, we conducted experiments using various combinations of modalities and loss functions to determine the optimal configuration for the brain-to-text model in this limited-data situation. More specifically, we utilize Brain Module Défossez et al. (2023) and a pre-trained whisper model Radford et al. (2023) to align brain representation in three aspects as shown in Figure 1, the Mel spectrogram, hidden state, and text. 1) We first align the Brain module with audio in the Mel spectrogram feature space to learn low-level features, such as acoustic features. 2) Secondly, we align the hidden state output in latent space from both whisper encoders of which input is predicted and ground truth Mel spectrogram respectively, enhancing the model's ability to extract high-level semantic features. 3) Lastly, we align the text representation from both streams within the framework.

Comprehensive experiments are conducted by utilizing non-invasive public MEG data from *GWilliams* Gwilliams et al. (2023) dataset, which captured MEG signals during a speech listening task. We have identified several noteworthy findings regarding brain-to-text alignment. 1) High-level semantic representations play a predominant role in MEG-to-text decoding, outperforming low-level acoustic features or direct text alignment strategies. 2) While low-level features contribute to improved performance when combined with high-level semantic information, they are insufficient in isolation to achieve satisfactory decoding results. 3) Explicit text alignment mechanisms prove detrimental to the task, significantly compromising the model's generalization capabilities. 4) Fine-tuning large-scale pre-trained models on this specialized, small-scale MEG dataset results in severe over-fitting, highlighting the challenges of transfer learning in this domain.

Remarkably, MAD **is capable of generalizing to unseen text**. Performance is evaluated using translation text relevancy metrics Papineni et al. (2002); Lin (2004). On raw MEG waves, MAD achieves 10.44 BLEU-1 on *GWilliams* **without teacher-forcing** evaluation on **entirely unseen text** which largely exceeds the current state-of-the-art (SOTA) performance. This paper also provides insights through numerous ablation studies to help people understand the impact of each component on aligning the MEG signal with texts. The contributions of this research could be summarized as follows:

- MAD presents an end-to-end neural network design for the direct conversion of MEG signals into text in open vocabulary, eliminating the dependence on word time segmentation provided by eye-tracker, teacher-forcing, or pretraining, representing the initial implementation of translating raw MEG waves into text for unseen content.

- We are the first to investigate various alignments and demonstrate the benefits of aligning with speech modality rather than text modality in the MEG-to-Text transcription task, offering significant insights for network improvement.

- Our extensive experimentation and thorough analysis of the proposed model showcase its effectiveness and highlight its superiority over existing methods in terms of translation accuracy, efficiency, and reliability.

## 2    RELATED WORKS

The discipline of converting brain signals into textual output has undergone considerable development in the contemporary era. In 2019, Anumanchipalli et al. Anumanchipalli et al. (2019) introduced a pioneering model capable of translating ECoG patterns into the articulatory movements necessary for speech production, subsequently generating acoustic properties such as MFCCs, leading to the production of intelligible speech. This landmark study ignited further exploration within the field. In the subsequent year, Wang et al. Wang et al. (2020) leveraged the capabilities of generative adversarial networks (GANs) to decipher ECoG data and synthesize speech. The year following, Willett et al. Willett et al. (2021) engineered a system that utilized a recurrent neural network (RNN) alongside a probabilistic language model to decode letters from neural activity during the act of handwriting. Most recently, Metzger et al. Metzger et al. (2022) constructed a sequence of processes that converted ECoG signals into textual information using an RNN, enhancing the results with the GPT-2 language model.

Within the domain of open-vocabulary interpretation, Metzger et al. Metzger et al. (2023) unveiled an RNN architecture capable of real-time decoding of speech, text, sentiment, and facial expressions from ECoG data. Simultaneously, Willett et al. Willett et al. (2023) managed to interpret text directly from neural activity. Liu et al. Liu et al. (2023) introduced a tripartite model designed to decode logo-syllabic languages, such as Chinese, by transforming ECoG signals into Chinese pinyin inclusive of tones and syllables, followed by speech synthesis. In a related development, Feng et al. Feng et al. (2023) achieved text interpretation from SEEG recordings. It is essential to highlight that these functional systems are predominantly reliant on invasive neural recordings.

In the domain of non-invasive neural recording, Meta unveiled a brain-to-speech system that leverages contrastive learning with MEG and EEG data Défossez et al. (2023). While this system is proficient in categorizing a constrained set of sentences, it is not conducive to open-vocabulary textual interpretation. Ghazaryan et al. Ghazaryan et al. (2023) explored the decoding of a restricted vocabulary from MEG responses. Wang et al. Wang & Ji (2022) crafted a mechanism for translating EEG features at the word level into text, employing a pretrained BART model Lewis et al. (2020). Subsequent investigations, including Dewave Duan et al. (2023), adopted the methodology established by Wang et al. Wang & Ji (2022), proposing a schema that incorporates wave2vec Baevski et al. (2020) and discrete codex for robust representations, which are subsequently funneled into a BART Lewis et al. (2020) model for text synthesis. These approaches, however, are dependent on teacher-forcing and disregard the necessity of comparing results with noise-injected inputs, potentially resulting in an inflated assessment of system efficacy. Recent scholarship Hyejeong et al. (2024) has revealed the limitations of these methods.

Yang et al. Yang et al. (2024) proposed an end-to-end paradigm for converting MEG signals to text, demonstrating high performance when training and evaluation sets were fully overlapped. Nevertheless, the model failed to demonstrate comparable performance when applied to unseen text. Our approach diverges from these methods by employing transfer learning with assistance of extra modality (Mel spectrogram) to align the model through multiple stages with low-level and high-level features of the ground truth. This enables our model to learn more effectively and generalize better to unseen text.

## 3    METHOD

### 3.1    TASK DEFINITION

Given a sequence of raw segment-level MEG signals $\varepsilon$, the goal is to decode the associated open-vocabulary text tokens $T$. This task also incorporates additional information in the form of speech $\Xi$. The MEG-Speech-Text pairs $\langle \varepsilon, \Xi, T \rangle$ are collected during speech perception. Our approach focuses on decoding $T$ using only the raw MEG signal $\varepsilon$, with the support of $\Xi$. MAD represents the first attempt at tackling this MEG to unseen text translation challenge.

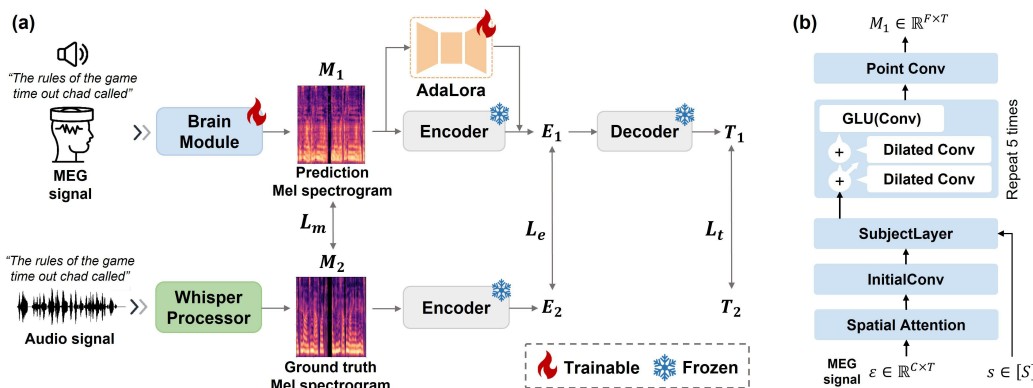

Figure 1. (a) Overview of our MAD architecture. We added alignments on the Mel spectrogram, the hidden states, and the text. There are three types of alignment, which are either based on our physics world (text and speech) or a largely pretrained model. $M_1$, $M_2$ is predicted and ground truth Mel spectrogram, $E_1$, $E_2$ is the hidden state of MEG-input and speech-input encoder respectively. $T_1$, and $T_2$ are predicted and ground truth text respectively. (b) Detailed architecture of the brain module Défossez et al. (2023). The MEG signal ($\varepsilon$ with $C$ recording channels and $T$ time points) is input to the brain module, and the output becomes the predicted Mel spectrogram ($M_1$ with $F$ features and $T$ time points). The selection of the 'Subject Layer' is determined by the subject index $s$.

## 3.2 ARCHITECTURE

Figure 1 presents an overview of our proposed model architecture. Our approach leverages transfer learning techniques to enhance performance on unseen text. We utilize the encoder and decoder models from the Whisper model Radford et al. (2023), a transformer-based encoder-decoder architecture known for robust speech recognition in challenging environments. We used AdaLoRA Zhang et al. (2023) module to train the whisper encoder in our architecture for saving memory.

The key innovation in our design lies in its multi-modal architecture, which is intentionally streamlined to facilitate experiments with different modalities. This design allows us to investigate which modalities contribute most effectively to the task, providing valuable insights into the MEG-to-Text decoding process.

Our architecture combines two primary modalities: speech and MEG signals. Both inputs are first converted to Mel spectrograms ($M_1$ for MEG, $M_2$ for speech) and then transformed into encoder features ($E_1$ and $E_2$ respectively). The speech input's features serve as the ground truth for the MEG-derived features. For the MEG input pathway, we further employ a decoder to predict text ($T_1$), using the actual transcription ($T_2$) as ground truth.

The Brain Module, adapted from Défossez et al. (2023), processes the MEG signals to Mel spectrograms. While we don't modify its internal architecture, its integration into our multi-modal framework is crucial for our experimental design. The brain module processes MEG signals through a deep neural network. It begins with a spatial attention layer, followed by a 1x1 convolution without activation. A subject-specific 'Subject Layer' is then selected using the subject index $s$, applying a 1x1 convolution unique to each participant. A residual dilated convolution block is applied, repeated five times as described in the paper Défossez et al. (2023). Each block consists of three convolutional layers. The first two dilated convolution layers in each block incorporate residual skip connections, while the third convolution layer uses a GLU (Gated Linear Unit) activation, which reduces the number of channels by half. And two 1x1 convolutions were applied. The final output is a predicted Mel spectrogram ($M_1$), which shares the same time points as the input MEG signals ($\varepsilon$). This modular approach allows us to systematically evaluate the contribution of each modality and their interactions, providing a flexible framework for exploring various alignment strategies in MEG-to-Text decoding.

## 3.3 LOSS

We employ three distinct loss functions tailored to align different modalities, each chosen for its established effectiveness in the respective context. The Mel spectrogram alignment loss, $L_m$, utilizes

CLIP loss Radford et al. (2021), which is particularly suited for this task due to its ability to learn a joint embedding space that enhances the representation of semantically related Mel spectrogram pairs. This is crucial as it allows the model to capture the nuances of MEG-speech relationships, thereby improving performance in multi-modal understanding. For the encoder model to effectively learn high-level features from the extracted representations, we adopt Maximum Mean Discrepancy (MMD) loss Borgwardt et al. (2006) as $L_e$. This choice stems from MMD's ability to measure and minimize the divergence between probability distributions, ensuring that the encoder's output aligns closely with the target distributions and facilitating better generalization across different domains. Finally, we implement cross-entropy loss, denoted as $L_t$, for the comparison between predicted and ground truth text. This loss is fundamental in classification tasks, as it quantifies the discrepancy between the predicted probabilities and actual labels, driving the model to refine its predictions iteratively. Collectively, these loss functions not only optimize the learning process for each modality but also foster an integrated approach to multi-modal learning that enhances overall model robustness and accuracy. The overall loss $L$ is below:

$$L = \lambda_m \cdot L_m + \lambda_e \cdot L_e + \lambda_t \cdot L_t, \tag{1}$$

where $L_m$ is $L_{\text{CLIP}}$, $L_E$ is $L_{\text{MMD}}$, $L_t$ is $L_{\text{CE}}$ in default.

The CLIP loss Radford et al. (2021) function originally operates on feature representations derived from both image and text modalities. It calculates similarity scores between these representations, with the objective of minimizing the distance between matching pairs while maximizing the distance between non-matching pairs. This framework enables the CLIP model to learn a joint embedding space, positioning semantically similar image-text pairs in close proximity. This capability facilitates tasks such as zero-shot image classification and text-based image retrieval. It has also been proved to be useful in predicting speech features as studies by Défossez et al. (2023). In our application, $L_m$ uses CLIP loss, which is applied on the Mel spectrogram in default. Mel spectrogram $M_1$ and $M_2$ is represented in three dimensions, we first flatten the batch size and time length dimensions to create a single dimension. The loss is then computed accordingly as follows:

---
**Algorithm 1:** CLIP-like Loss Calculation

**Input:** $M_1 [n, d_m]$ Predicted Mel spectrogram ,
$M_2 [n, d_m]$ Ground truth Mel spectrogram ,
$d_m$ Dimensionality of multimodal embedding,
$t$ Learned temperature parameter,
$n$ Batch size.
**Output:** CLIP loss
1 $logits \leftarrow M_1 \cdot M_2^T \cdot e^t$ ; // Scaled pairwise cosine similarities, [n,n]
2 $labels \leftarrow \text{Range}(n)$ ;                        // Labels for each example
3 $loss_1 \leftarrow \text{CrossEntropyLoss}(logits, labels, \text{axis} = 0)$;
4 $loss_2 \leftarrow \text{CrossEntropyLoss}(logits, labels, \text{axis} = 1)$;
5 $L_m \leftarrow \text{Mean}(loss_1, loss_2)$;
6 **return** $L_{CLIP}$;

---

The MMD loss (Maximum Mean Discrepancy loss) Borgwardt et al. (2006) is a measure of the discrepancy between two probability distributions. It is commonly used in domain adaptation and generative modeling to encourage the distributions of source and target data to be similar. If we flatten the hidden state $E_1$ and $E_2$ of the batch size $N$, time dimension $T_d$ and feature dimension $D_e$, it will run out of memory if we input full length into the model, so we randomly select features time-wise of length $T_r$, therefore the selected features is $E_r$ shape is $[N, T_r, D_e]$. Here, the function $\phi$ represents a kernel mapping that projects the original variables into a Reproducing Kernel Hilbert Space (RKHS). This kernel is crucial for defining the similarity between the distributions in the context of MMD. The formula for the MMD loss is:

$$L_{\text{MMD}} = \sum_{t=1}^{T_r} \frac{1}{n} \left\| \sum_{i=1}^{n} \phi(E_{1r}(i,t)) - \sum_{i=1}^{n} \phi(E_{2r}(i,t)) \right\|_{\mathcal{H}}, \tag{2}$$

where $E_{1r}(i,t)$ and $E_{2r}(i,t)$ represent the feature vector at the $i$-th sample and $t$-th time step from the randomly selected features $E_{1r}$ and $E_{2r}$ respectively. The notation $\| \cdot \|_{\mathcal{H}}$ denotes the norm in the RKHS induced by the kernel function $\phi$.

For an Automatic Speech Recognition (ASR) system, the cross-entropy (CE) loss is commonly used as a loss function to train the model. $T_1$, $T_2$ are predicted and ground truth text tokens. Here we define $N$ as batch size, $J$ as token length and $C$ as number of output classes in the language head. The CE loss in the context of ASR can be defined as follows:

$$L_{\text{CE}} = -\frac{1}{N} \sum_{n=1}^{N} \sum_{j=1}^{J} \sum_{c=1}^{C} T_{1,n,j,c} \log(T_{2,n,j,c}). \tag{3}$$

## 4 EXPERIMENTS

### 4.1 DATASET

The GWilliams dataset Gwilliams et al. (2023) is a magnetoencephalography (MEG) dataset designed for assessing natural speech comprehension. It features authentic MEG recordings from 27 participants proficient in English. These participants engaged in two separate sessions, each involving two hours of listening to four stories, which are "cable spool fort","easy money","lw1","the black willow". To get a fair evaluation, we split our dataset directly on stories, we test on "cable spool fort", validate on "lw1" and train on other stories. Details are in Table. 1. For more details about the dataset, please refer to Supp. B.

Table 1. Details about the dataset splits, we ensured the three splits are totally separated. Unique sentences means the sentences that are different with other sentences, same meaning for unique words. There is no overlap sentence between train and test set. 371(46%) means 371 words in test set is also in train set, accounting for 46 percentage.

| Split | Segments | Unique sentences | Words | Unique words | Overlap sentence | Overlap words |
|---|---|---|---|---|---|---|
| train | 133966 | 13266 | 150497 | 2776 | - | - |
| validation | 14896 | 1387 | 156027 | 478 | - | - |
| test | 31115 | 3151 | 355654 | 805 | 0 | 371(46%) |

For preprocessing, we used first band pass filter the MEG signal $\varepsilon$ between 1 Hz and 40 Hz, then it is resampled to 100Hz to reduce computing. We ensure that we separated training, evaluation, testing set totally since we used one story for testing, another story for evaluation, last two ones for training. We extract 4-second windows from the MEG-speech-text pairs, sliding every second and randomly shifting the window by ±0.5 seconds to generate samples. Speech $\Xi$ is then transformed to Mel $M$ with window length of 400, hop length of 160, which is the original configuration in Whisper model Radford et al. (2023), since the setted speech sampling rate is 16kHz, after conversion, $M$ is of shape [400, 80] time and feature wise for 4 second speech, then it is matched with $\varepsilon$ of time length 400.

### 4.2 IMPLEMENTATION DETAILS

All models were trained using Nvidia 4090 (24GB) GPUs. Training was conducted with a learning rate of 3e-4 and a batch size of 32 over 5 epochs, selecting the best-performing model based on evaluation loss. AdamW was employed as the optimizer across all models. Each experiment takes about 18 hours on signal GPU with 8 workers to finish. Lambda value in all experiment on MAD model set as follows: $\lambda_m = 1$, $\lambda_e = 0.01$, $\lambda_t = 1$.

### 4.3 EVALUATION METRICS

The performance comparison of our proposed MAD model with other state-of-the-art models is summarized in Table 2. The table highlights various configurations and the corresponding evaluation metrics, 1) BLEU-1Papineni et al. (2002): Assesses the accuracy of machine-translated text. 2) ROUGE-1Lin (2004): Measures the quality of automatic summarization. 3) BertScoreZhang et al. (2019): Evaluates semantic similarity. 4) CERMartins & Garland Jr (1991): Measures the accuracy of speech recognition. 5) Self-BLEU Zhu et al. (2018): Assesses the diversity of generated text.

Table 2. Comparison with other models. Lo is LoRA, B is brain module. Bert here means Bertscore. Results is obtained without teacher forcing in evaluation. Here, Tr stands for trainable modules. B-1 stands for BLEU-1. R-1 stands for ROUGE-1-F. SB stands for Self-BLEU. RS means randomly selecting sentences from test set as predictions. As we can see, only MAD is much higher than RS on BLEU-1 score.

| Modality | Method | Tr | Loss | B-1(%)↑ | R-1 (%)↑ | Bert(%)↑ | CER(%)↓ | SB(%)↓ |
|---|---|---|---|---|---|---|---|---|
| - | RS | - | - | 5.86 | 7.20 | 83.73 | 87.30 | 96.12 |
| MEG | NeuSpeech Yang et al. (2024) | Lo | $L_t$ | 5.49 | 8.43 | 83.98 | 77.02 | 99.7 |
| MEG | Wav2vec2CTC Défossez et al. (2023) | B | $L_m$ | 0.55 | 1.44 | 76.02 | 152.23 | 92.67 |
| MEG | MAD | B | $L_m + L_e$ | 10.44 | 6.93 | 83.39 | 89.82 | 85.66 |
| Noise | MAD | B | $L_m + L_e$ | 3.87 | 3.16 | 83.20 | 126.95 | 87.54 |
| MEG | MAD w/tf | B | $L_m + L_e$ | 12.93 | 18.28 | 82.87 | 74.31 | 83.35 |
| Noise | MAD w/tf | B | $L_m + L_e$ | 0.19 | 6.68 | 59.92 | 87.57 | 68.63 |

We compare the performance of our proposed model, MAD, against existing state-of-the-art methods, NeuSpeech Yang et al. (2024), Wav2vec2CTC Défossez et al. (2023) for decoding MEG signals into text. Besides, we compared our results to random selecting and input noise as two effective baselines to show the performance lower bound.

NeuSpeech Yang et al. (2024) is an encoder-decoder framework model used for MEG, utilizing the Low-Rank Adaptation (LoRA) method with a text-based loss ($L_t$), achieves best scores on ROUGE-1-F, BertScore, and CER. However, the self-bleu score is almost 100%, which means the generation always repeat same thing. Besides, the BLEU-1 score is lower than RS, which means these three metrics are not reliable, which is further discussed in Supp. C.

Wav2vec2CTC Défossez et al. (2023): The original model predicts the output of the Wav2vec2 Baevski et al. (2020) encoder with brain module. We add the pretrained language model head in the Wav2vec2CTC Baevski et al. (2020) model as another baseline. This model shows significantly lower performance across all metrics, which is not effective.

Our MAD model, which integrates the brain module with a combined loss ($L_m + L_e$), demonstrates superior performance with a BLEU-1 score of 10.44% which is about 5 points higher than NeuSpeech Yang et al. (2024) and RS. Besides, we compared the performance of our model when it receives pure Gaussian noise which is the shape of the MEG signal to show that our model is generating text based on MEG signal. For noise input, MAD's performance BLEU-1 dropped to 3.87%, indicating that MAD model has learned from the MEG signal rather than just noise. Additionally, we evaluated MAD with teacher-forcing. When teacher-forcing was applied (MAD w/tf), the model's performance significantly improved, achieving a BLEU-1 score of 12.93% and a ROUGE-1-F score of 18.28%, confirming the effectiveness of teacher-forcing in enhancing model performance. Similarly, the BLEU-1 score for noise w/tf is low too (0.19%), further indicating our model can distinguish noise and MEG. In addition, our model has low Self-BLEU which means our model is generate diverse sentences according to MEG signal rather than simply repeating.

Overall, our MAD model achieved SOTA performance for MEG-to-Text decoding compared to previous SOTA models, demonstrating significant progress in MEG-to-Text translation. Additionally, we performed a fair comparison with noise and RS, which served as two error bars to validate the robustness and reliability of our model's performance. Furthermore, the self-BLEU scores indicated the diversity of our model's generated text, demonstrating its ability to truly learn and generalize from the data. Next section, we will show the generated sample along with the Mel spectrogram to further show the effectiveness of our MAD model.

### 4.4 GENERATED SAMPLES

#### 4.4.1 TEXT

Table 3 showcases the performance of our proposed MAD model compared to NeuSpeechYang et al. (2024) and Wav2vecCTCDéfossez et al. (2023) in the challenging task of MEG-to-Text decoding. The results clearly demonstrate MAD's superiority in multiple aspects.

MAD exhibits exceptional semantic capture capabilities, particularly without teacher-forcing. It generates words that directly match the ground truth, such as "step", "in", and "eyes", across various

Table 3. Transcription results. These are some results obtained without teacher forcing evaluation. **Bold** for exact matched words, *italy* for similar semantic or pronunciation words. w/ tf means with teacher forcing in evaluation. We lower case results of Wav2vecCTC to give a better visual experience.

| Decoding Results on *GWilliams* Gwilliams et al. (2023) |
| --- |
| Ground Truth: in one hand and the screwdriver held up high in the other ready to step down into |
| MAD: As **to the** worst folk, we are a **step in** his *floor* **in** it **to** separate from prepanded time |
| MAD w/ tf: of **one** *otherdriver* **the to**. **the** front **hand to** flip **up**. **the** |
| NeuSpeech: He looked at me **and** said **to** me, |
| NeuSpeech w/ tf: he **the** of. **the other** was **the**.. **the** middle.. take on. |
| Wav2vecCTC: hoas whoistd ban hes hoe leingd s woe stoind hae score mend chroa |
| Ground Truth: expression and crossed eyes, the tumbleweed in one hand and the |
| MAD: Primarized. Ribid **the** fire is *closed*. Your **eyes** to **the** thumps |
| MAD w/ tf: followed **the eyes** found **the** of, **the** other. **in** |
| NeuSpeech: He looked at me **and** said to me, |
| NeuSpeech w/ tf: heired. **the the**. he wordsult, of **the**'s, **the** |
| Wav2vecCTC: hien scroucst oin hs oarcsthoins hoer li's b |
| Ground Truth: the awesomeness of what he intended pulling his eyes |
| MAD: your **eyes** panned out your **eyes** clear **eye** pain |
| MAD w/ tf: *esomeess* **the** is has to **the eyes** to |
| NeuSpeech: **He** *looked* at me and said, I'm not sure **what**'s going on. |
| NeuSpeech w/ tf: **he** wayestomeess of **the** he had to. fingers. his |
| Wav2vecCTC: is thoane horalaug lind hes schoragthrascre d scrond sfhoanxs s |

examples. Moreover, MAD produces semantically related phrases like "step in his floor in", which strongly correlates with the ground truth "step down into", showcasing MAD's capacity to capture not just individual words but broader semantic concepts.

Notably, even with teacher-forcing, MAD maintains strong semantic relevance. It accurately identifies key words like "one", "hand", "up", and "eyes", and even approximates complex words such as "screwdriver" (generated as "otherdriver"). This is particularly significant as it highlights MAD's robustness across different decoding strategies.

In contrast, NeuSpeech, both with and without teacher-forcing, struggles to capture specific semantic content. Without teacher-forcing, it repetitively generates generic sentences like "He looked at me and said to me," showing little variation across different inputs. When teacher-forcing is applied, NeuSpeech's output, while more varied, lacks the semantic accuracy demonstrated by MAD. For instance, it fails to consistently produce relevant nouns or maintain context, unlike MAD which successfully identifies key terms across various scenarios. Wav2vecCTC consistently produces phonetically-based outputs that lack coherence and relevance to the target sentences, falling short of the meaningful content generation achieved by MAD.

In conclusion, our proposed MAD model represents a significant advancement in MEG-to-Text decoding. It consistently outperforms existing approaches in semantic relevance, word-level accuracy, and concept capture, showcasing a deeper understanding of the complex relationship between MEG signals and natural language. MAD's robust performance across different decoding strategies sets a new standard in the field, paving the way for more accurate and context-aware MEG-based text generation systems.

### 4.4.2 MEL SPECTROGRAM

More than text, we showed the Mel spectrogram in Figure 2. It presents the Mel spectrogram of the two sample sentences in the test set. In this context, it is employed to compare the predicted audio signal generated by the model with the actual ground truth audio signal in the form of Mel spectrogram.

Upon examining the spectrograms of two samples, several observations can be made regarding the model's capabilities and performance. 1) There is a general similarity between prediction and ground truth in the overall structure, 2) the model learns some fine-grained details such as temporal variations in the low-frequency regions which have bigger energy than the high-frequency region, 3) the model

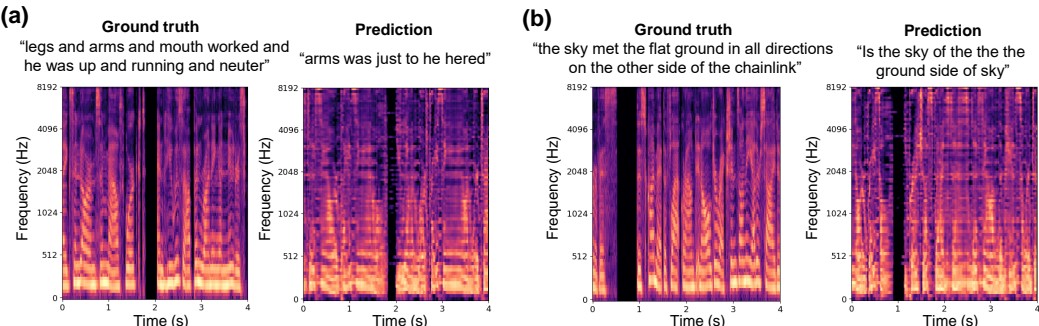

Figure 2. Two sample examples from the test set: (a) and (b) represent different samples. Ground truth refers to Mel spectrograms of the audio signal processed by the whisper processor. Predictions refer to Mel spectrograms generated by the brain module. The predicted text was generated using teacher-forcing.

can predict the speech signal's temporal blanks, proving it understands the MEG features associated with the absence of speech. However, significant discrepancies are apparent. While the ground truth spectrogram displays a more complex and detailed pattern with distinct frequency bands and variations over time, the predicted spectrogram seems less detailed and exhibits more uniform and repetitive patterns.

These discrepancies highlight the current limitations of the model in producing high-quality, accurate, natural audio signals from MEG data. Future work can introduce pretrained generative models in speech modality to improve the model's ability to learn and represent these fine-grained details, which is important for accurate speech recognition.

## 4.5 MODEL ABLATION

We conducted three ablation studies to evaluate our model's effectiveness and robustness in MEG-to-Text decoding. These studies are designed to assess different aspects of the model; 1) Systematic analysis of the impact of model architecture and loss functions, 2) Evaluation of model performance in subject-dependent scenarios, and 3) Evaluation of model robustness under various noise conditions. 1) is provided here and the results of 2), 3) are discussed in Appendix A.

Table 4. Performance of the MAD model across different trainable components and loss functions. Where B and Lo denote the brain module and LoRA applied to the encoder, respectively. These results are obtained **without** teacher forcing in evaluation. Be default, $L_m$ is CLIP loss, $L_e$ is MMD loss, () means loss type replacement. B-1 is the abbreviation of BLEU-1. R-1 is the ROUGE-1-F. SB is self-BLEU. The direction of arrow on metrics indicates better text decoding performance

| Loss | Trainables | B-1 (%)↑ | R-1 (%)↑ | Bert (%)↑ | CER (%)↓ | SB (%)↓ |
|---|---|---|---|---|---|---|
| $L_m$ | B | 1.88 | 2.24 | 79.83 | 83.65 | 99.03 |
| $L_e$ | B | 10.09 | 6.29 | 82.74 | 88.84 | 83.62 |
| $L_e + L_t$ | B | 6.15 | 4.81 | 84.43 | 80.33 | 95.32 |
| $L_m + L_e(\text{CLIP})$ | B | 2.04 | 1.14 | 81.91 | 94.85 | 96.16 |
| $L_m(\text{MMD}) + L_e$ | B | 9.64 | 5.71 | 81.62 | 87.95 | 80.55 |
| $L_m + L_e$ | B | **10.44** | 6.93 | 83.39 | 89.82 | 85.28 |
| $L_m + L_e + L_t$ | B | 7.14 | 4.37 | 82.29 | 88.40 | 83.95 |
| $L_m + L_e$ | B+Lo | 1.13 | 0.79 | 81.17 | 87.65 | 99.98 |
| $L_m + L_e + L_t$ | B+Lo | 8.33 | 6.40 | 83.14 | 91.43 | 99.11 |

Table 4 presents a comparison of various configurations including different combinations of loss functions, loss types and trainable modules, which reveals several crucial insights that may pave the

way for more effective and generalizable approaches to brain2text tasks in the context of limited specialized data:

1. High-level feature alignment ($L_e$) proves critical for model performance in MEG-to-Text conversion. When used as a single loss function, $L_e$ achieves a BLEU-1 score of 10.09, which is remarkably close to the highest score of 10.44 obtained with combined losses. This demonstrates the crucial role of aligning high-level semantic features in facilitating accurate mapping from brain activity patterns to linguistic constructs.

2. Low-level features alignment ($L_m$) can complement high-level semantic features ($L_e$) to some extent, marginally improving performance when combined appropriately. However, they are ineffective when used in isolation. The addition of $L_m$ to $L_e$ slightly increases the BLEU-1 score from 10.09 to 10.44, and similarly, adding $L_m$ to $L_e + L_t$ improves performance. Conversely, when $L_m$ is used alone, it yields a notably low BLEU-1 score of 1.88, underscoring its limited efficacy as a standalone loss function in this task.

3. Text alignment ($L_t$) proves detrimental to model performance in this brain-to-text task. When $L_t$ is added to $L_e$ and $L_m + L_e$, BLEU-1 performance decreases approximately 3 points, with self-BLEU escalating sharply to over 95%. This counterintuitive finding suggests that explicit text reconstruction may interfere with the model's ability to generalize effectively from MEG signals to text in limited data scenarios.

4. Introducing LoRA as trainable parameters leads to severe overfitting, with self-BLEU scores exceeding 99%. This suggests a mismatch between large-scale pretraining and fine-tuning on limited MEG data, cautioning against direct fine-tuning of large pretrained models on small, specialized datasets.

## 5 LIMITATION

Although our MAD model outperforms previous SOTA models, we have to point out that this model's generation is far from practical utilization in reality since the performance is much lower than speech recognition models. Besides, this work is implemented on listening datasets, which is different from silent speech.

## 6 CONCLUSION

In this paper, we presented MAD, a novel end-to-end training framework for MEG-to-Text translation. Our model leverages multiple alignment utilizing auxiliary modalities, which aligns brain activity data more effectively with corresponding textual outputs. Experimental results suggest that the newly proposed MAD framework achieves 10.44 BLEU-1 on *GWilliams* **without teacher-forcing** evaluation on **entirely unseen text**, significantly surpassing the current state-of-the-art performance.

Through comprehensive ablation studies, we share valuable insights into the efficacy of our approach, designing better loss function combinations to inspire future research. Our findings highlight the importance of high-level semantic alignment, the complementary role of low-level features, and the potential pitfalls of explicit text reconstruction and over-reliance on large pretrained models. These insights underscore the potential of the MAD framework in neural decoding, offering a robust strategy for MEG-to-Text translation by effectively capturing complex patterns in MEG signals and translating them into coherent text, which can be used by later-on research.

In conclusion, our proposed MAD framework significantly advances the state-of-the-art(SOTA) in MEG-to-text decoding, offering new avenues for enhancing communication tools for individuals with severe speech and motor impairments. This work sets the stage for further exploration into multi-modal alignments and their impact on neural decoding systems. Future research can focus on refining alignment mechanisms, exploring more sophisticated feature integration techniques, and extending the application of our model to more diverse linguistic tasks and larger-scale MEG datasets.

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

# SUPPLEMENTARY MATERIAL FOR
# MAD: MULTI-ALIGNMENT MEG-TO-TEXT DECODING

## A  DISCUSSION

### A.1  EVALUATION OF MODEL PERFORMANCE IN SUBJECT-DEPENDENT SCENARIO

Most previous studies have employed subject-independent scenarios. However, due to the significant individual variation in biological signals, this approach may not be optimal for real-world applications Thulasidas et al. (2006). Therefore, we also conducted experiments under subject-dependent scenarios. We randomly held out three subjects from the train set and tested the performance across the three subjects in Table. 5. Among the three subjects, B-1 scores ranged from 8.09 to 10.33, with comparable performance observed across other metrics such as R-1, BERT, CER, and SB scores. This demonstrates the robustness of our model in overcoming the issue of subject dependency.

Table 5. Metrics on 3 random selected subject id of 7, 15, 26. We train and validate on data that strictly excluded these three subjects, then we test on each subject. The number attached to each point is the performance of the metric of each axis.

|     | subject id | B-1 | R-1 | Bert | CER | SB |
|-----|-----------|-----|-----|------|-----|----|
| MEG | 7 | 10.33 | 6.85 | 83.67 | 88.40 | 86.93 |
| MEG | 15 | 8.09 | 5.76 | 84.34 | 86.13 | 85.56 |
| MEG | 26 | 10.60 | 7.34 | 82.98 | 92.22 | 87.84 |

### A.2  EVALUATION OF MODEL ROBUSTNESS UNDER VARIOUS NOISE CONDITIONS

Table 6. Different noise sample strategies on MAD performance. Noise is standard gaussian, Shuffle channel is switching channels randomly, shuffle time is shuffling timestamps within each channel, Channelwise Gaussion is sampling gaussian noise with mean and standard deviation within each channel, Timewise Gaussian is sampling gaussian noise with mean and standard deviation within each timestamp.

| Input | Method | B-1 | R-1 | Bert | CER | SB |
|-------|--------|-----|-----|------|-----|----|
| MEG | MAD | 10.44 | 6.93 | 83.39 | 89.82 | 85.66 |
| Noise | MAD | 3.87 | 3.16 | 83.20 | 126.95 | 87.54 |
| Shuffle channel | MAD | 6.13 | 4.96 | 84.16 | 88.90 | 87.07 |
| Shuffle time | MAD | 5.57 | 4.85 | 83.28 | 82.86 | 91.24 |
| Channel wise Gaussian | MAD | 5.81 | 5.00 | 83.34 | 82.97 | 88.54 |
| Timewise Gaussian | MAD | 5.41 | 4.79 | 83.31 | 83.21 | 89.50 |

Table. 6 shows results of different noise sample strategies to prove our model is robust against four different types of noise. These results show that disrupting the temporal or channel information in the MEG signal leads to a significant drop in performance, as reflected by the lower B-1 scores ranging from 5.41 to 6.13. This underscores the robustness of the original MEG data and suggests that our model is fine-tuned to the precise temporal and channel-wise information in MEG signals.

Table. 7 demonstrates the effect of gradually increasing noise portions on model performance, as measured by the B-1 score. As the proportion of noise increases, the performance decreases consistently. This suggests that our model is highly sensitive to noise, and its ability to interpret the MEG signals degrades as the noise level rises.

## B  DATASET

The Gwilliams Gwilliams et al. (2023) dataset is described below:

Table 7. This table illustrates the method of injecting noise into the input signal using the formula $output = input \cdot (1 - a) + noise \cdot a$, where $a$ denotes the proportion of noise, $input$ represents the original signal, and $noise$ is a noise signal with the same mean and standard deviation as the input signal within each channel.

| Noise ratio (%) | 100 | 90 | 80 | 70 | 60 | 50 | 40 | 30 | 20 | 10 | 0 |
|---|---|---|---|---|---|---|---|---|---|---|---|
| B-1 | 5.81 | 5.65 | 5.16 | 5.21 | 6.21 | 7.5 | 8.57 | 9.29 | 9.87 | 10.19 | 10.44 |

## B.1 PARTICIPANTS

- **Total Participants:** 27 English-speaking adults (15 females)
- **Age:** Mean = 24.8 years, SD = 6.4 years
- **Recruitment:** Subject pool of NYU Abu Dhabi
- **Consent and Compensation:** All provided written informed consent and were compensated
- **Health:** Reported normal hearing and no history of neurological disorders
- **Language:** All but one participant (S20) were native English speakers
- **Sessions:**
    - Majority (22 participants) performed two identical one-hour-long sessions
    - Sessions were separated by 1 day to 2 months
- **Ethics Approval:** Approved by the IRB ethics committee of NYU Abu Dhabi

## B.2 PROCEDURE

- **Recording Sessions:**
    - Duration: Each session lasted approximately 1 hour.
    - Equipment: Recorded with a 208 axial-gradiometer MEG scanner (Kanazawa Institute of Technology).
    - Sampling Rate: 1,000 Hz.
    - Filtering: Online band-pass filtered between 0.01 and 200 Hz.
    - Task: Participants listened to four distinct stories through binaural tube earphones (Aero Technologies) at a mean level of 70 dB sound pressure level.
- **Pre-Experiment Exposure:**
    - Participants were exposed to 20 seconds of each distinct speaker voice.
    - Purpose: To clarify session structure and familiarize participants with the voices.
- **Story Presentation Order:**
    - Assigned pseudo-randomly using a "Latin-square design."
    - Same order used for both recording sessions for each participant.
- **Attention Check:**
    - Participants answered a two-alternative forced-choice question every 3 minutes.
    - Example Question: "What precious material had Chuck found? Diamonds or Gold."
    - Average Accuracy: 98%, confirming engagement and comprehension.
- **MRI Scans:**
    - T1-weighted anatomical scans were performed after MEG recording if not already available.
    - Six participants did not return for their T1 scan.
- **Head Shape Digitization:**
    - Head shape digitized with a hand-held FastSCAN laser scanner (Polhemus).
    - Co-registered with five head-position coils.
    - Coil positions collected before and after each recording, stored in the 'marker' file.
    - Experimenter continuously monitored head position to minimize movement.

### B.3 STIMULI

- **Stories:** Four English fictional stories selected from the Open American National Corpus:
  - **'Cable spool boy':** 1,948 words, narrating two young brothers playing in the woods.
  - **'LW1':** 861 words, narrating an alien spaceship trying to find its way home.
  - **'Black willow':** 4,652 words, narrating the difficulties an author encounters during writing.
  - **'Easy money':** 3,541 words, narrating two friends using a magical trick to make money.
- **Audio Tracks:**
  - Synthesized using Mac OS Mojave's (c) text-to-speech.
  - Voices and speech rates varied every 5-20 sentences to decorrelate language from acoustic representations.
  - Voices used: 'Ava', 'Samantha', and 'Allison'.
  - Speech rate: Between 145 and 205 words per minute.
  - Silence between sentences: Varied between 0 and 1,000 ms.
- **Story Segments:**
  - Each story divided into 5-minute sound files.
  - Random word list played approximately every 30 seconds, generated from unique content words of the preceding segment.
  - Very small fraction (<1%) of non-words introduced in natural sentences.
- **Task Definition:**
  - Each "task" corresponds to the concatenation of sentences and word lists.
  - All subjects listened to the same set of four tasks, in different block orders.

## C  DISCUSSION ABOUT THE MAIN TABLE

We used BLEU-1 Papineni et al. (2002),ROUGE-1-F Lin (2004),BertScore Zhang et al. (2019),CER Martins & Garland Jr (1991),Self-BLEU Zhu et al. (2018) as metrics in the main table to show the capability of previous models and our models. However, as observed, NeuSpeech Yang et al. (2024) model has the best score for ROUGE-1,Bert,CER, which is incredible, therefore we measured the Self-BLEU of this model, which is almost 100%, and found out NeuSpeech predicts almost the same sentence "He looked at me and said to me" all the time for different sentences in Supp. 1. Generation of this bad quality has best score on these three metrics, which means these three metrics are not effective in measuring the generation quality. Therefore, we think BLEU-1 the most reliable metric in this task for now. Besides, we randomly selected sentences, which is RS in the table, from the test set as another baseline, we found out that the BLEU-1 score is higher than NeuSpeech, which means the NeuSpeech model is not effective, which is very reasonable. After all, it seems that using BLEU score is the only reasonable metric of evaluating the quality of generated text.

As observed in the table, it is very clear that our MAD model is significantly higher than RS and NeuSpeech and Wav2vec2CTC on BLEU-1, which means our MAD model is effective on unseen text.

## D  MORE GENERATED SAMPLES

We showed more generate samples here.

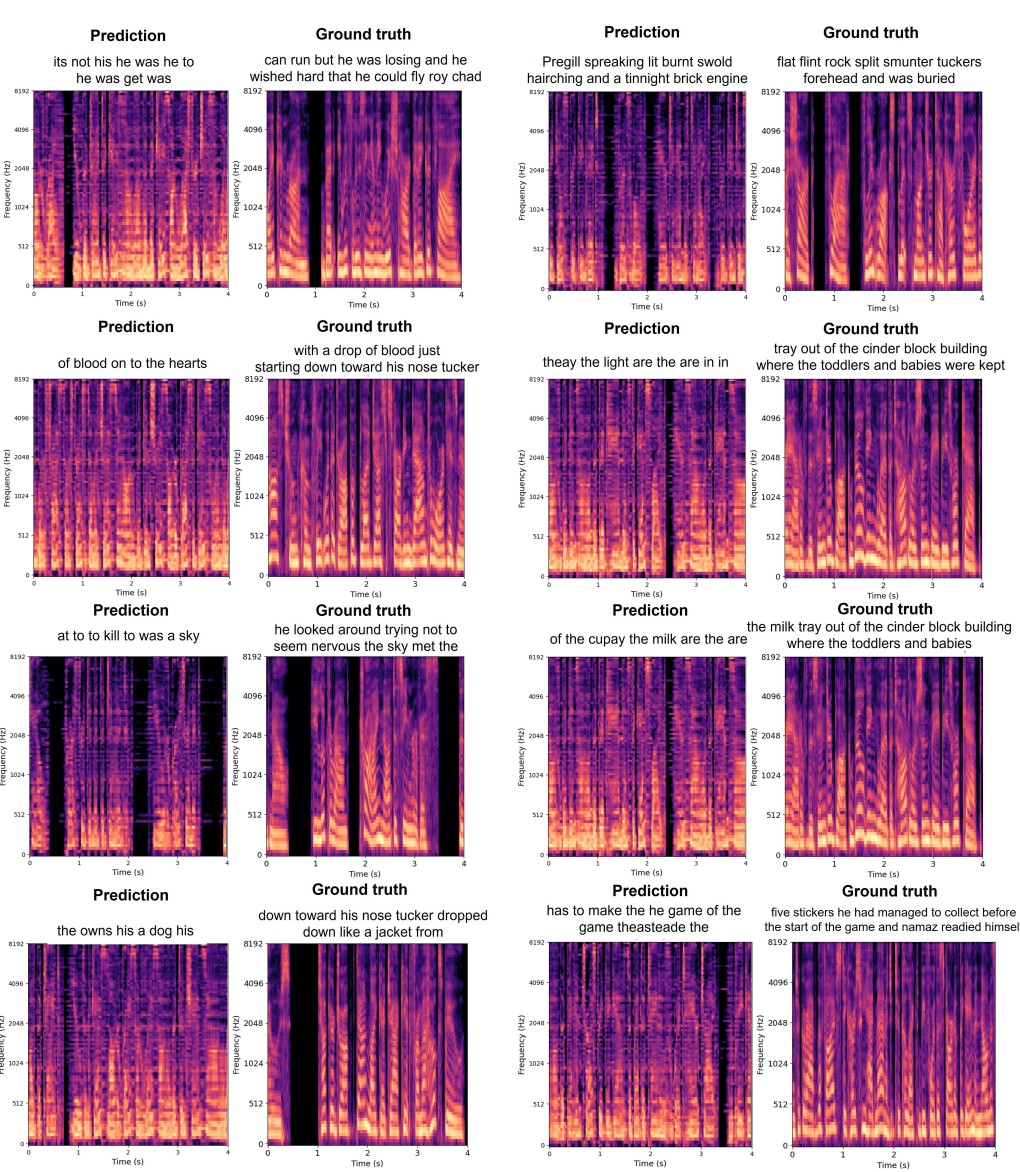

Figure 3. Eight sample examples of the test set. Prediction refers to Mel spectrograms generated by the brain module. Ground truth refers to Mel spectrograms of the audio signal processed by the whisper processor. The predicted text was generated using teacher-forcing. These examples were produced using $L_m(mmd)$ with only a trainable brain module.

Listing 1. NeuSpeech Yang et al. (2024) generation without teacher-forcing.

```
1  start*******************************
2  Predicted:  He looked at me and said to me,
3  True: were smelly thistles or cocklebur stems covered with spiked
4  end==================================
5
6  start*******************************
7  Predicted:  He looked at me and said to me,
8  True: or ordering Chad around or something. But since his fall the year
       before,
9  end==================================
10
11 start*******************************
12 Predicted:  I'm not sure how to do it. It's just a little bit more
13 True: oldest boy in the playground, and the one who decided the rules
14 end==================================
15
16 start*******************************
17 Predicted:  He looked at me and said to me,
18 True: Spauw for fear of what was coming next. I'll make you fight. Tucker
19 end==================================
20
21 start*******************************
22 Predicted:  he looked at me and said, I don't know what to do.
23 True: before, Roy had been shuffling and doing what he was told. Chad
24 end==================================
25
26 start*******************************
27 Predicted:  He looked at me and said to me,
28 True: for the tumbleweed to prove he wasn't a baby to Tucker. But as much
29 end==================================
30
31 start*******************************
32 Predicted:  He looked at me and said to me,
33 True: walk really every something great blade over. Mama
34 end==================================
35
36 start*******************************
37 Predicted:  He looked at me and said to me,
38 True: other ready to step down into Chad's back. A sharp, Flat,
39 end==================================
40
41 start*******************************
42 Predicted:  He looked at me and said to me,
43 True: about gathering stickers himself. Roy was too
44 end==================================
45
46 start*******************************
47 Predicted:  He looked at me and said to me,
48 True: in shade and napped inside the walls. Then could wild and blink-
       breath corner-hard
49 end==================================
```

Listing 2. Wav2vec2CTC Défossez et al. (2023) generation.

```
start********************************
Predicted: THLE'S HOAN BSFBHLAG'DS HON CITES HAG THOEANGLEN S QJRANGD
    HOAND'S SORUESTHO E MRERLWOAINS HOAX TH
True: AND NAPPED INSIDE THE WALLS THEN COULD WILD AND BLINKBREATH
    CORNERHARD
end=================================

start********************************
Predicted: SHROE BHOING TSEDTRAINS BBB
True: OF TIRES TWO BIG TRACTOR TIRES CAPPED OUT WITH ONE FROM A TRUCK AND
     TWO SMALLER
end=================================

start********************************
Predicted: IES HO BHE HRORA SCIRCIND FBW
True: THAT OUT EITHER IT WAS ROY'S FAVORITE GAME NO
end=================================

start********************************
Predicted: AGSCHRONDSOUNE HIRS ON HOIN PHRORLI'S HEXSHIS B
True: ABOUT GATHERING STICKERS HIMSELF ROY WAS
end=================================

start********************************
Predicted: D JABWUISD BHOEND TE AUST THORE MLADS BHAXTS BMOIST OND F
True: TWO SMALLER ONES FROM CARS THE OLDER BOYS LAY AROUND IN
end=================================

start********************************
Predicted: CHORWALDES OE CSCRER BXSCOUE WONSTFBHE HOITS PR ENS
True: WASN'T CHICKEN YOU WANNA PLAY ROBOTS ROY ASKED CHAD
end=================================

start********************************
Predicted: BHI'S JMA
True: WHAT YOU SUCK CHAD SAID HE WISHED ROY
end=================================

start********************************
Predicted: SHOUDTIES BVIEN HOAS S
True: MAKING A DOOR TO THE SMALL ROOM INSIDE THE TALL TUMBLEWEED FLAG
end=================================

start********************************
Predicted: IDH HOASTD HIE' SCHORK SPHRERG 'S THOANS OABLWSDT'T XSCIED
    HRIE HOER SPTHRALNINDSFOFTHES PHE CHOR HIER
True: WEAPONS ALLOWED ACCORDING TO HUMPTY DUMPTY NURSERY RULES
end=================================

start********************************
Predicted: SHOURX PHRERLNGDS FHOANS OMBLWSDT'T ESCED RIE HORN
    SFTHRANINDSFOTS FHE CHOR CHIRE HINS HIND HOURXS TH
True: ALLOWED ACCORDING TO HUMPTY DUMPTY NURSERY RULES OF ENGAGEMENT
end=================================
```

Listing 3. MAD generation with teacher-forcing.

```
start********************************
Predicted: orus said wast be a day but out
True: chad said he wished roy wouldnt fall for that gag every time get
end===================================

start********************************
Predicted: name is from his head on his head ofs
True: down his head rose and his eyes focused over chads shoulder out roy
end===================================

start********************************
Predicted: be the smell times have at
True: until he could smell the dust several hated must staring brother
end===================================

start********************************
Predicted: he had not but though he was not a be down to
True: he wished he were there now even if he did have to sit next
end===================================

start********************************
Predicted: is sky the the the ground side of the sky
True: the sky met the flat ground in all directions on the other side of
    the chainlink fence
end===================================

start********************************
Predicted: the the up lift him know the the rest the that ist the fool
    the he
True: to lift him and let him reach for the tumbleweed to prove he wasnt
    a baby to tucker but
end===================================

start********************************
Predicted: sound is the mouth ist been
True: a sick sound but the thing in his head hadnt worked
end===================================

start********************************
Predicted: the of the top a red medal
True: out of the top of the black fort like a gold headed monster
end===================================

start********************************
Predicted: the roy him  he name was be
True: out after him roy chad called but his voice would
end===================================

start********************************
Predicted: soldiers astronautss a on be us and
True: for soldiers and astronauts and its vote going to help roy
end===================================
```

