# OpenReview forum: "MAD: Multi-Alignment MEG-to-Text Decoding"
_ICLR.cc/2025/Conference — ICLR 2025 Conference Withdrawn Submission_

### Official Review · Reviewer_A4Q8 · 2024-10-25

**Soundness:** 3
**Presentation:** 3
**Contribution:** 3
**Rating:** 6
**Confidence:** 4

**Summary:**

The paper introduces MAD, an end-to-end multi-alignment framework for translating MEG signals to text. This framework addresses challenges related to decoding unseen text by aligning MEG data with speech and text modalities. Using the GWilliams dataset, MAD achieves improved performance in MEG-to-text generation compared to prior approaches. The study also examines the impact of different loss functions on decoding performance, offering insights that may aid in future optimization of MEG-based language decoding.

**Strengths:**

- This paper introduces a new open-vocabulary MEG-to-Text translation framework MAD, avoiding the teacher-forcing problem seen in previous models and using a more reasonable evaluation method.

- The experimental design is robust, including comprehensive ablation studies that underscore the advantages of multi-modal alignment.

- Explanations of the model architecture, data, and experimental setup are clear and well-structured.

- The model sets a new standard in MEG-to-Text decoding accuracy, showcasing the valuable potential for real-world BCI applications.

**Weaknesses:**

The primary concern is the limited dataset, as the experiments were only conducted on the GWilliams dataset, while the model was trained with MEG-Speech-Text modality pairs. Testing on a single dataset limits the model's robustness and generalizability claims.

**Questions:**

1.	Lack of Multi-Dataset Testing: The model is only evaluated on the GWilliams dataset, unlike related works such as NeuSpeech and Wav2vecCTC, which test on additional datasets like Schoffelen. Although the authors acknowledge risks of overfitting on small datasets and discuss generalizability, a practical demonstration on other datasets like Schoffelen would better validate the model’s transferability and robustness.
2.	Inference Process for Sliding Windows: The model uses 4-second sliding windows with a shift of 1 second, incorporating random shifts of ±0.5 seconds to generate training samples. However, it remains unclear whether the inference stage aggregates outputs across windows or applies additional processing steps. Moreover, this raises concerns about the effect of different window sizes on the decoding results, an aspect not explored in the paper.
3.	Ablation Study on Loss Functions: The ablation studies discuss three loss functions—high-level feature alignment ($L_e$), low-level feature alignment ($L_m$), and text alignment ($L_t$)—but do not provide concrete evidence of how each influences decoding results. Adding examples of decoding results under different combinations of these loss functions would provide valuable insight, especially in validating the importance of high-level and low-level feature alignments as argued by the authors.

---

### Official Review · Reviewer_4LRw · 2024-10-28

**Soundness:** 2
**Presentation:** 3
**Contribution:** 3
**Rating:** 5
**Confidence:** 3

**Summary:**

This paper proposes a novel brain-to-text perceived speech decoding pipeline (MAD) that leverages alignments from multiple modalities. The key contributions highlighted by the authors are the ability of their framework to generalize to unseen text, eliminating the need for word time segmentation (e.g. teacher-forcing, pre-training, or via eye-tracker), and a performance analysis with a suite of metrics, ablation testing, and benchmarking against other models. A comparison of different modalities is possible due to the modular structure of their design.

**Strengths:**

The approach as a whole is novel and the reported performance is strong across metrics for both high level accuracy and low level semantic content. The inclusion of random gaussian baselines should be noted and adopted as standard for future decoding studies. The push for evaluations on unseen text is also laudable.

**Weaknesses:**

There appears to be some issue with citation formatting that should be fixed (see ICLR 2025 style guide), as well as a few grammatical issues. Additionally, the benchmarking comparison with Défossez et al. (2023) appears to be in bad faith. Instead of comparing the performance of their decoding framework against Défossez et al.'s model as originally designed, it seems as if the authors of this paper use only the brain model and then apply a decoding head in the style of their framework. Thus, it results in many of the generated unigrams being gibberish as opposed to actual words and greatly reduces whatever the real performance of the Défossez et al. model should be across all reported metrics. The original model from Défossez et al. returns the most likely segment of audio given the meg input and therefore always returns real words. The paper by Défossez et al. explicitly mentions the ability to decode segments not present in the training set. Thus, it is still reasonable to benchmark with the Défossez et al. model as originally designed. If the intention was to underscore that the model from Défossez et al. needs access to the segmented test audio while the proposed framework operates with MEG data alone, this could have been stated and the Défossez et al. model omitted from bechmarking comparison.

**Questions:**

Questions:
- Are there results of benchmarking of the proposed model against other SOTA models using different test splits (i.e. not for entirely unseen data)? or with the Défossez et al. model as originally proposed (i.e. classification with the unseen test audio)?

Suggestions:
- fix the citation styling (see section 4.1 of the Formatting Instructions for ICLR 2025 Conference Submissions)

---

### Official Review · Reviewer_kWaa · 2024-10-30

**Soundness:** 2
**Presentation:** 2
**Contribution:** 1
**Rating:** 3
**Confidence:** 4

**Summary:**

The submission studies text decoding from brain activity based on a publicly available dataset that contains magnetoencephalography (MEG) recording of 27 subjects while they were passively listening to different stories.
Building upon recent prior work, the authors utilize a brain module and train it to predict mel spectrograms derived from the audio signals.
These MEG-derived mel spectrograms are then fed through a publicly available, pre-trained speech decoding model (whisper, OpenAI) to complete the MEG-to-text decoder.
The authors propose a combination of three loss terms with the aim to align mel spectrograms, latent speech encoder representations as well as decoder outputs and labels.
Given the small amount of data, they find that training merely the brain module parameters and freezing the remaining modules yielded best results (BLEU-1 of 10.44 % without teacher forcing) if the model was trained to align the latent speech encoder representations and the mel spectrograms.
Although the proposed method outperforms the considered baseline models, the overall poor performance (BLEU-1 values of approx. 10%) prevents practical utilization.

**Strengths:**

The paper can be classified as a combination of existing ideas.
In terms of architecture, the authors follow the idea of (Yang et al. 2024) and combine a convolutional network with a pre-trained whisper speech decoder; instead of standard convolution layers they propose to use the brain module of (Défossez et al. 2023).
As training objectives, they compare combinations of similar (and related) loss functions as proposed in (Défossez et al. 2023) and (Yang et al. 2024).

### Quality

I appreciate that the authors evaluated the models on unseen stories and put substantial effort in assessing the significance of the results. They compare the obtained results against shuffled data and models with random inputs.

### Clarity

The problem setting and the overall approach of the contribution are clearly communicated.

### Significance

The work confirms that pre-trained speech models (e.g., whisper here or wave2vec in (Défossez et al. 2023)) are helpful to extract a significant amount of language information from non-invasive brain activity (MEG here). Yet, the reported generalization results to unseen stories (Table 2) and subjects (Table 5) are discouraging.

### References

A. Défossez, C. Caucheteux, J. Rapin, O. Kabeli, and J.-R. King, “Decoding speech perception from non-invasive brain recordings,” Nat Mach Intell, Oct. 2023, doi: 10.1038/s42256-023-00714-5.

Y. Yang et al., “NeuSpeech: Decode Neural signal as Speech,” Jun. 03, 2024, arXiv: 2403.01748.[Online]. Available: http://arxiv.org/abs/2403.01748

**Weaknesses:**

### Originality
- Brain activity (invasive and non-invasive) to speech/text translation has already been introduced in prior works, as summarized in section 2 (related work).

- The methodological contribution is incremental.
Défossez et al. (2023) introduced the idea of aligning M/EEG signals with latent representations of a pre-trained ASR model (wav2vec2.0) with the CLIP loss. In the submission, the authors use a related approach (whisper encoder instead of wav2vce2.0, and MMD loss instead of CLIP).
Yang et al. (2024) combined the whisper model with a convolutional adapter network (trained with AdaLoRa). In this submission, the authors propose a modified architecture that uses the brain module of (Défossez et al. 2023) as adapter network.

### Quality
- While I appreciate that the authors report results of the considered performance metrics for random effects (i.e., random shuffling and noise as input), their approach is insufficient to actually estimate the significance of the results. Instead of a single evaluation they should have used permutation testing (see e.g., Maris 2012) to estimate the metrics' distribution under the null hypothesis (i.e., no-relation between the MEG-derived text and the ground truth).

- Unlike (Défossez et al. 2023) and (Yang et al. 2024) the authors decided to analyze only one public dataset. I think the authors should have also analyzed the results for the (Schoffelen et al. 2019) dataset. Beyond that, the results lack quantification of the stability/variability of the results across random weight initializations.

### Clarity
- Figure 1(a) does not correspond to the proposed MAD model in Table 2.
This discrepancy suggests that the study might be affected by the double dipping problem (Kriegeskorte et al. 2009) in the sense that the authors tested many different configurations and then picked the final MAD model based on the test set result.

### References

A. Défossez, C. Caucheteux, J. Rapin, O. Kabeli, and J.-R. King, “Decoding speech perception from non-invasive brain recordings,” Nat Mach Intell, Oct. 2023, doi: 10.1038/s42256-023-00714-5.

Y. Yang et al., “NeuSpeech: Decode Neural signal as Speech,” Jun. 03, 2024, arXiv: 2403.01748.[Online]. Available: http://arxiv.org/abs/2403.01748

J.-M. Schoffelen, R. Oostenveld, N. H. L. Lam, J. Uddén, A. Hultén, and P. Hagoort, “A 204-subject multimodal neuroimaging dataset to study language processing,” Sci Data, vol. 6, no. 1, p. 17, Apr. 2019, doi: 10.1038/s41597-019-0020-y.

E. Maris, “Statistical testing in electrophysiological studies,” Psychophysiology, vol. 49, no. 4, pp. 549–565, 2012, doi: 10.1111/j.1469-8986.2011.01320.x.

N. Kriegeskorte, W. K. Simmons, P. S. F. Bellgowan, and C. I. Baker, “Circular analysis in systems neuroscience: the dangers of double dipping,” Nat Neurosci, vol. 12, no. 5, pp. 535–540, May 2009, doi: 10.1038/nn.2303.

**Questions:**

### Methods
- Please use consistent symbols for the same concept (for example either $n$ or $N$ for the batch size)
- $\phi$ is not defined in (2)
- line 304: define the random shifts. Were they samples from the interval $[-0.5, 0.5]$ or the set $\{-0.5, 0.5\}$?
- the considered baseline method `Wav2vec2CTC` was not proposed in (Défossez et al. 2023).


### Wording, Grammar and Organization
- citation formatting: please use the `\citep` and `\citet` latex commands appropriately.
- line 42: "letter" -> "letters"
- line 201: "don't" -> "do not"
- lines 207 to 208: grammar issues
- line 231: "in default" -> "by default"
- some references miss important information (e.g., the journal). Please check thoroughly.


### References

A. Défossez, C. Caucheteux, J. Rapin, O. Kabeli, and J.-R. King, “Decoding speech perception from non-invasive brain recordings,” Nat Mach Intell, Oct. 2023, doi: 10.1038/s42256-023-00714-5.

---

> ### Author Response · Authors · 2024-11-26
>
> I want to get some advice from you since you seem to be very professional. I want to ask some questions:
> 1. about permutation testing, what I need to do to implement the test is to shuffle the order of ground truth sentences and maintain the order of predicted sentences? Then, if the score is much lower after this shuffle, this means the model is significant? Is this testing method widely accepted by everyone in this field, so that no one can doubt the effectiveness of the model after passing this permutation test?
> 2. how high do you think the BLEU score should be to be practical? If the performance is still very low, but it passed the permutation testing, should the method be considered as practical, or a stepstone to be practical?
> 3. should we not compare the Brainimagick model proposed by A. Défossez et al.? since they are intended to decode text, but to match audio segment. Do you think it is OK to not compare with this model which is denoted as Wav2vecCTC in our paper.
>
> Thank you so much for your suggestions and advice.

---

> > ### Comment · Reviewer_kWaa · 2024-11-29
> >
> > > 1. about permutation testing, what I need to do to implement the test is to shuffle the order of ground truth sentences and maintain the order of predicted sentences? Then, if the score is much lower after this shuffle, this means the model is significant? Is this testing method widely accepted by everyone in this field, so that no one can doubt the effectiveness of the model after passing this permutation test?
> >
> > If you do permutation testing, you need to estimate the distribution under the null hypothesis. This can be done via shuffling repeatedly (ideally 1,000 to 10,000 times). Each time you get a summary statistic (e.g., BLEU score, accuracy, ...). After many iterations you can reasonable well estimate the quantiles of your null distribution. If you accept 5% type 1 error, the 95% quantile should be your threshold. If your observed summary statistic is higher than that value, then you can reject the null hypothesis (with a remaining 5% risk of making a type 1 error).
> >
> > For further details, please study the paper [1] that I shared earlier, or any standard textbook on statistical testing with permutation tests.
> >
> > [1] E. Maris, “Statistical testing in electrophysiological studies,” Psychophysiology, vol. 49, no. 4, pp. 549–565, 2012, doi: 10.1111/j.1469-8986.2011.01320.x.
> >
> > > 2. how high do you think the BLEU score should be to be practical? If the performance is still very low, but it passed the permutation testing, should the method be considered as practical, or a stepstone to be practical?
> >
> > Practical relevance depends on the application field. At this stage it is important to convincingly demonstrate that the method performs better than a random model. Additionally, the gap to BLEU scores obtained with invasive techniques should be small.
> >
> > > 3. should we not compare the Brainimagick model proposed by A. Défossez et al.? since they are intended to decode text, but to match audio segment. Do you think it is OK to not compare with this model which is denoted as Wav2vecCTC in our paper.
> >
> > I think it is okay to keep the comparison. However, the text in the submitted manuscript reads as if (Défossez et al. 2023) proposed Wav2vecCTC. This is not true. You should clearly state that you used some ideas/modules from (Défossez et al. 2023) and combined them with a CTC loss (and maybe more changes) so that you have an additional baseline method  for your problem setting.

---

### Official Review · Reviewer_4z7h · 2024-10-31

**Soundness:** 2
**Presentation:** 2
**Contribution:** 2
**Rating:** 3
**Confidence:** 5

**Summary:**

This paper proposes a MEG2Text framework, which consists of a Brain Module and a Whisper Codec. MEG is initially decoded into a Mel spectrogram and subsequently translated into text using the Whisper model. The framework incorporates three alignments: low-level, high-level, and text-level alignments, which have potential inspiration for neural transcription.

**Strengths:**

1. The incorporation of a new modality in the MEG2Text translation is desirable.
2. High-level L_e loss is proven to be effective.

**Weaknesses:**

The residual connection in Figure 1 (b) is missing an arrow.
	The main experimental results in Table 2 have no advantage over RS and NeuSpeech, and are almost all lower except for B-1 and self-B.
	The brain module within Wav2vecCTC is not trained on text, so it exhibits poor performance and struggles to generate coherent words. The comparison is unfair.
	Baseline models should be compared both w/ and w/o tf.
	The paper aims to showcase the model's ability to generalize to unseen text, yet this attribute is not evident within the experimental setup presented. The paper solely conducts experiments on a test set with 46% overlap in words, which fails to represent unseen text. For a more thorough evaluation, it is crucial to compare the model with the baselines on both seen and unseen text separately. Furthermore, experiments should be conducted on additional datasets to enhance the validation of generalization.
	Ablation studies have demonstrated that L_t and LoRA are redundant and should be omitted from the model structure.

**Questions:**

1.	What is the purpose of random window shifting in line 304?
2.	How is cross-subject implemented in Section A.1? Why does it perform similarly to intra-subject despite having a different subject index input to the brain module?

---

> ### Author Response · Authors · 2024-11-26
>
> Thank you for your previous time and valuable advice. We will be very happy if you can provide some advice for us.
>
> A1. About comparison with Wav2vecCTC :
>
> We actually trained Wav2vecCTC with text, the performance is poor too, it cannot generate coherent words as well, but to compare with Brainimagick which does not use text in training, we only showed the results by training this without using text. Do you think we should not denote this method as proposed by A. Défossez et al. ? Instead, we just compare the Wav2vecCTC model which is trained with text?
>
> A2, About unseen text:
>
> Unseen text does not necessarily imply that every single word within it is entirely new or unfamiliar to the reader or the model. The concept of “unseen text” refers to a combination of words or a sentence structure that has not been encountered before. Just as Shakespeare and Bob Dylan may have used some of the same words in their works, this does not mean that Bob Dylan’s lyrics are derivative or previously encountered simply because they share common vocabulary with Shakespeare’s texts.
>
> Each author brings a unique style, context, and message to their work, and the same words can take on entirely different meanings when placed in a different context or arrangement. The novelty of a text lies not just in the individual words, but in the original combination and the fresh perspective that the author brings to those words. Therefore, even if individual words are recognizable, the unseen text is characterized by its unique assembly and the new ideas or emotions it conveys, which sets it apart from any previously encountered material.

---

### Official Review · Reviewer_jPuN · 2024-10-31

**Soundness:** 3
**Presentation:** 3
**Contribution:** 3
**Rating:** 6
**Confidence:** 4

**Summary:**

This paper presents **MAD**, an end-to-end framework for decoding MEG signals into text. They use a novel multi-alignment approach with audio and text representations. They present their results utilizing the GWilliams dataset showing high BLEU score on entirely unseen text. They highlight the framework's potential for generalization. They conduct comprehensive ablation studies that highlight the important of high dimensional representations alignment over direct text alignment for robust performance.

**Strengths:**

- **Clarity**: The paper is exceptionally well-written and easy to understand.
- **Novelty**: It introduces a novel multi-alignment approach that utilizes auxiliary modalities, such as Mel spectrograms, to enhance the translation of MEG signals into text. This work is notable for being one of the few that reports results without using teacher forcing.
- **Performance**: The method achieves state-of-the-art results on the BLEU-1 metric for this dataset.
- **Ablation Studies**: The paper includes comprehensive experiments and ablation studies that clarify the contributions of each model component, especially the loss functions and alignment mechanisms.

**Weaknesses:**

- **Dataset**: The reliance on a single dataset reduces the impact of the results. Although the authors claim good performance on entirely unseen text, the word overlap is 46% on stories belonging to the same corpus and same subjects. This assumption should be validated across other datasets.

- **Losses**: The authors indicate that the Le loss is the most important in the study, but they do not explain why. A part-of-speech analysis could clarify the performance differences between content and function words. They assert that high-level speech is better decoded than low-level speech; however, this has been demonstrated in previous studies (https://www.biorxiv.org/content/10.1101/2024.04.19.590280v1). Additionally, the authors do not provide a solid hypothesis for why the Lt loss negatively impacts the BLEU-1 score. They also do not show the performance when only using Lt, as they do for Lm and Le.

- **Noise Study**: In the Evaluation Metrics section, noise is employed to demonstrate that the model effectively learns from MEG signals. It would be more insightful to train the model on noise rather than simply evaluating it.

- **Metrics**: Although the paper mentions five different metrics, the discussion predominantly focuses on the BLEU-1 score, while the ROUGE-1 and CER scores exhibit different (opposite?) behaviors that do not show SOTA results. Notably, one of the main findings regarding the detrimental effect of Lt is not corroborated by the CER metrics. While it is commendable to use five metrics, the authors primarily discuss BLEU-1 without explaining why it is the most relevant metric.

- **Remarks**:
  - **Introduction and Related Work**: Consider integrating the related work section into the introduction for better flow, or relocating some introductory elements to the related work section.
  - **Dataset Description**: The paper states 2 hours per session, but it should be corrected to 1 hour per session, as per the MEG-MASC paper (Methods/Participants section, https://www.nature.com/articles/s41597-023-02752-5).
  - **Limitations**: The limitations section is underdeveloped and should be expanded.
  - **Typos**: Correct "setted" to "set" (section 4.1), "primarized" to "prioritized" (section 4.4.1), add a missing space between "state-of-the-art" and "(SOTA)" (conclusion), and address incorrect references (Supplementary A1).
  - **References**: On two different computers, the PDF displays a "double" reference in the Related Work section, mentioning the same studies twice in succession.

**Questions:**

- **Dataset**: Could you clarify your choice of dataset? Why did you select GWilliams instead of alternatives like Armeni (https://www.nature.com/articles/s41597-022-01382-7)? If the model is SOTA on unseen text, why not test it on other available datasets?

- **Inputs**: Can you explain how you transform a shifted 4-second window into a fixed-size sample for the Mel spectrogram?

- **Splitting**: Would it be possible to include experiments that demonstrate performance using different data splits (i.e., different combinations of the 4 stories)?

- **Multi-Modality Integration**: Could you elaborate on the challenges encountered when aligning MEG data with Mel spectrograms and hidden states?

- **Losses**: Can you provide the performance results when only using the Lt loss? Additionally, could you expand on the significance of the Le loss in this setup?

- **Discussion**: Could you elaborate on the other metrics used in your analysis? How does the performance behavior differ among them, and what insights do they provide to enhance the overall analysis?

- **Noise**: Would you consider experimenting with a model trained on noise, rather than merely evaluating it on noisy data?

---

### Note · Authors · 2024-12-02

I have read and agree with the venue's withdrawal policy on behalf of myself and my co-authors.